# A Novel Finger Vein Recognition Method Based on Aggregation of Radon-Like Features

**DOI:** 10.3390/s21051885

**Published:** 2021-03-08

**Authors:** Qiong Yao, Dan Song, Xiang Xu, Kun Zou

**Affiliations:** Artificial Intelligence and Computer Vision Laboratory, University of Electronic Science and Technology of China, Zhongshan Institute, Zhongshan 528402, China; yaoqiong@zsc.edu.cn (Q.Y.); songdan@zsc.edu.cn (D.S.); cszoukun@zsc.edu.cn (K.Z.)

**Keywords:** finger vein, biometrics, mean curvature, radon-like features

## Abstract

Finger vein (FV) biometrics is one of the most promising individual recognition traits, which has the capabilities of uniqueness, anti-forgery, and bio-assay, etc. However, due to the restricts of imaging environments, the acquired FV images are easily degraded to low-contrast, blur, as well as serious noise disturbance. Therefore, how to extract more efficient and robust features from these low-quality FV images, remains to be addressed. In this paper, a novel feature extraction method of FV images is presented, which combines curvature and radon-like features (RLF). First, an enhanced vein pattern image is obtained by calculating the mean curvature of each pixel in the original FV image. Then, a specific implementation of RLF is developed and performed on the previously obtained vein pattern image, which can effectively aggregate the dispersed spatial information around the vein structures, thus highlight vein patterns and suppress spurious non-boundary responses and noises. Finally, a smoother vein structure image is obtained for subsequent matching and verification. Compared with the existing curvature-based recognition methods, the proposed method can not only preserve the inherent vein patterns, but also eliminate most of the pseudo vein information, so as to restore more smoothing and genuine vein structure information. In order to assess the performance of our proposed RLF-based method, we conducted comprehensive experiments on three public FV databases and a self-built FV database (which contains 37,080 samples that derived from 1030 individuals). The experimental results denoted that RLF-based feature extraction method can obtain more complete and continuous vein patterns, as well as better recognition accuracy.

## 1. Introduction

Finger vein (FV) biometrics is an efficient individual recognition trait, which has the advantages of uniqueness, anti-forgery, bio-assay, permanence, and user-friendly [1,2,3]. At present, the authentication technologies based on FV traits have shown wide application prospects in the fields of airports, banks, consumer electronics, and so on [4,5]. However, since the FV images are usually acquired under the restricted imaging environments, not only do imaging types of equipment need to be designed as narrow and compact as possible, but the illuminations of infrared light are often weak and uneven, leading to the acquired images appear to low-contrast, blur, and noisy. In this regard, how to extract more efficient and robust features from those low-quality images is particularly critical for the FV recognition system.

Generally speaking, feature extraction of FV images can be carried out in two ways (as shown in Figure 1). One way is to see an FV image as a general digital image, thereby, some mature feature extraction algorithms in the field of digital image processing can be directly migrated to use. In this case, features are extracted from the whole image while not distinguishing the vein and background (hereinafter we named as ‘image-level’ feature extraction). However, FV images have their own characteristics, for instance, the vein points are relatively sparse, and the variations of gray value between the veins and the surrounding background are slow and gradual. Thereby, the second way is to try to separate the vein patterns from the image, and then extract features on the pure vein patterns (hereinafter we named as ‘vein-level’ feature extraction). In essence, ‘vein-level’ methods obey the hypothesis that the vein points are generally darker than their surrounding non-vein points.

For the class of ‘image-level’ feature extraction, existing methods were mainly derived from the fields of face recognition [6,7], image classification [8,9], and remote sensing images [10], etc. Among these, local binary pattern (LBP) [11] and its variant methods, such as local line binary pattern (LLBP) [12], local derivative pattern (LDP) [13], local directional code (LDC) [14], personalized best bit map (PBBM) [15], personalized best patches map (PBPM) [16], discriminative binary code (DBC) [17,18], anchor-based manifold binary pattern (AMBP) [19], block multi-scale uniform local binary pattern (MULBP) [20], etc., have demonstrated satisfactory recognition performance. LBP-based operators transferred the whole FV image into an ordered set of binary values, which can be seen as a type of local statistical-based method. While different from LBP-based methods, competitive coding-based method [21] encoded an FV image according to certain rules. More specifically, only orientations of the minimal Gabor filter responses (that means the trend of lines) were encoded. As a result, the competitive coding-based method demonstrated insensitivity to illumination and better recognition accuracy. Besides, two statistical analysis methods based on subspace learning, principal component analysis (PCA) [22] and linear discriminant analysis (LDA) [23], were also introduced for FV image processing. PCA adopted an unsupervised linear transformation to obtain a set of orthogonal vectors with the largest variances, while LDA adopted a supervised transformation to obtain a discriminative subspace. Moreover, an ${\left(2D\right)}^{2}$ PCA technology was specially designed to extract two-orientational features [24]. In this manner, the process of converting a two-dimensional image into a one-dimensional vector is avoided. In addition to the aforementioned methods, superpixel-based methods [25] also belong to this class.

Although the ‘image-level’ methods avoid the process of separating vein patterns, they still have some drawbacks: First and for most, when the original FV images are low-quality, the local similarity of vein points and their surrounding non-vein points is high, which makes it difficult to strip out the vein points. In addition, because the genuine vein points in FV images are relatively sparse, a large amount of irrelevant and pseudo information are mixed as vein information, which hinders the matching performance.

Considering that each finger has its unique and consistent vein information, most methods of ‘vein-level’ class are devoted to separate more accurate vein patterns from the image. Among, line-shape-based and curvature-based methods are two representative branches. In order to extract line-shape of the veins, a repeated line tracking method was proposed in [26]. Later, a wide line detector (WLD) was proposed in [27], which considered width information of the veins. Likewise, curvature-based methods were also widely used for the representation of vein patterns. Mathematically, curvature reflects how much a curve bends at a certain point. Taking the FV image for example, on each cross-sectional profile, the maximum curvature points are those points that own the local minimum gray value [28]. After that, in [29], by decomposing the Hessian matrix of the normalized gradient image, two orthogonal principal curvatures of each point were calculated, and the larger one, which denotes the maximum curvature among all directions, was used to characterize the vein structures. In [30], mean curvature was utilized to trace the valley-like structures in a two-dimensional space. Recently, difference curvature with its greater capability in distinguishing edges and ramps, was also applied to extract vein features [31]. Roughly speaking, mean curvature and difference curvature both belong to two-dimensional curvature operators, while the maximum curvature belongs to one-dimensional curvature operator.

The ‘vein-level’ features represent the intrinsic vein patterns, which are intended to minimize the influence of non-vein information. However, many methods of this class still focus on solving problems from the perspective of each individual pixel, while neglecting the benefits of spatial correlation, thus leading to the sensitivity to weak intensity variations, and easy to generate many noises and irregular shadings in the obtained feature images.

In recent years, deep learning (DL) based methods, due to their ability of high-level feature learning, have also been introduced for FV image recognition [32]. Generally, DL-based models provide an end-to-end recognition procedure, and directly output the final matching results. Initially, researchers tend to design a few lightweight network architectures [33,34]. It is due to the fact that, on the one hand, training samples are always insufficient in some publicly available FV image databases; on the other hand, FV images contain relatively simpler semantic features (mainly line-shape features). In [35], a four-layer convolutional neural network (CNN), with two fused convolution/subsampling layers and two full connection layers, was constructed for FV recognition. Then, a light CNN, which integrated a *maxout* activation function [36] and a triple similarity loss function [37], was proposed in [38]. In [39], a lightweight two-stream CNN architecture was proposed for FV verification. Among, the first stream network was used to process original image pairs as input, and the second stream network was used to process mini-ROI pairs as input. Then, the outputs of two streams were concatenated to form the final feature representation.

Besides, some existing DL models, such as VGGNet [38,40,41,42], ResNet [43], and AlexNet [44], etc., were also introduced. In these models, either a different image or an image pair was fed into the networks. It should be noted that in some networks, the low-level features were used as inputs, e.g., line-shape features extracted by using WLD operator [27] were fed into a modified VGGNet-16 [41], thus promoting better recognition accuracy. Such idea of using low-level features also emerged in [45], in which an assemble feature extractor was constructed to integrate multiple low-level FV features, and then used to automatically pre-label the vein and background samples, so as to efficiently solve the problem of insufficient training samples.

Recently, some more powerful but complex network models, such as Siamese Network [46], GaborPCA Network [47,48], Convolutional Autoencoder [49], Capsule Network [50], DenseNet [51,52], Fully Convolutional Network (FCN) [53,54], Generative Adversarial Network (GAN) [55,56,57], and Long Short-term Memory (LSTM) Network [58], etc., also emerged in the field of FV image recognition. Especially for the GAN, which can not only achieve more robust vein patterns from low-quality FV images, but also generate a variety of synthetic FV samples.

Although DL-based FV recognition methods have achieved promising performance, they still suffer from some problems to be in suspense. First, DL-based models are all data-driven, in which the training data sources play an important role. However, most benchmark FV databases are small-scale, thus easy to bring overfitting problems. Secondly, in a real scenario, the FV images often have poor quality (blur, deformation, etc.), and many mature DL models require a resized input image, which will degrade the recognition performance. Therefore, how to extract effective and robust vein structures while removing the pseudo vein information as far as possible, will be a benefit to the DL-based methods. Third, in real-time processing, many DL-based models have heavy computation and huge hyper-parameters, which are hard to be ignored.

Inspired by the aforementioned methods, we presented a novel feature extraction method of FV images, which combined curvature and radon-like features (RLF) [59]. First, an enhanced vein pattern image was obtained by calculating the mean curvature of each pixel in the original FV image. However, due to the low quality of the original region of interest (ROI) image, the obtained vein pattern image not only contained geometric information of each vein point, but also distributed a lot of pseudo points with similar geometric information. At this point, if we do binarization directly, it will be bound to introduce more errors. So, we developed a specific implementation of RLF, and applied to the previously obtained vein pattern image, which can effectively aggregate the dispersed spatial information around the vein structures, thus highlight vein patterns and suppress spurious non-boundary responses and noises. Finally, a greater smoothing vein structure image was obtained for subsequent matching and verification.

The main idea of our proposed method is to realize a more advanced feature representation of FV images, which takes the existing local feature as an initial base-feature representation. Then, by means of spatial correlation, this kind of base-feature is reorganized and processed to form a more advanced feature. To be specific, in order to extract more clear vein patterns from low-quality FV images, we introduced the RLF [59] to aggregate the mean curvature-based features. The RLF has been successfully applied to the enhancement and segmentation of cell boundaries in connectomics. To the best of our knowledge, it is the first attempt to introduce RLF for feature representation of FV images. We have compared our proposed method with some commonly used methods, including LLBP [12], Gabor filters [60], WLD [27], as well as curvature-based methods [28,30], and confirmed that our method significantly outperforms the compared methods in the case of FV recognition. In summary, the main innovative contributions of our work are three folds:First and foremost, we present a novel feature representation method of FV images, which can be used to carry out spatial aggregation and feature refinement on the noisy vein pattern images, thus obtaining more robust vein structural information.Second, we develop a specific implementation of RLF, and apply for FV image processing. Compared with some commonly used feature extraction methods of FV images, our proposed RLF-based method can highlight vein patterns and suppress spurious non-boundary responses and noises, thus obtaining more smoothing vein structure images.Third, the implemented RLF-based feature extraction method demonstrates a fast running speed and a relatively low complexity of the algorithm. The experimental results also confirm the effectiveness of our method.

The remainder of this paper is organized as follows. Section 2 provides a brief review of the related works, including two key issues of mean curvature and radon-like features. Section 3 details our proposed RLF-based feature extraction method. Section 4 discusses the experimental results obtained by using four different FV databases. Section 5 concludes the paper with some remarks and hints at plausible future research lines.

## 2. Related Works

In this section, we briefly review the basic principle of two important issues in our proposed method, including mean curvature and radon-like features.

### 2.1. Mean Curvature

The concept of mean curvature was first put forward by Marie-Sophie Germain [61], which is defined by the arithmetic mean of any two orthogonal curvatures that are perpendicular to each other on a surface. Supposing two orthogonal curvatures are expressed as $\kappa_{1}$ and $\kappa_{2}$, the mean curvature will be calculated by $\bar{\kappa }=\left(\kappa_{1}+\kappa_{2}\right)/2$. Compared with maximum curvature, the mean curvature is calculated in a two-dimensional space, and according to Euler’s formula, it actually represents the average value of curvatures in all directions, so it is insensitive to orientation.

In the field of FV recognition, mean curvature was first adopted in [30]. Here, we use divergence of the normalized gradient vector to calculate the mean curvature values of each point. Meanwhile, for the two orthogonal directions, we directly select *x* and *y* axes for convenience. The corresponding formula is shown in Equation (Equation 1).
(1)$$\begin{array}{cc} \bar{\kappa } & =-\frac{1}{2}\nabla \cdot g\\ \; & =\frac{1}{2}\left(\frac{\partial g}{\partial x}+\frac{\partial g}{\partial y}\right)\\ \; & =\frac{1}{2}\left(\frac{I_{xx}I_{y}^{2}-2I_{xy}I_{x}I_{y}+I_{yy}I_{x}^{2}}{{\left(I_{x}^{2}+I_{y}^{2}\right)}^{3/2}}\right) \end{array}$$
where *I* denotes the image intensity field, g=∇I/|∇I| denotes the normalized gradient of image, $I_{x}$ and $I_{y}$ are two partial derivatives of the first order, while $I_{xx}$, $I_{xy}$ and $I_{yy}$ are partial derivatives of the second order. Equation (Equation 1) denotes that the mean curvature provides a quantitative measurement of the likeness degree to ridges or valleys, which is large at ridge-like structures and small at valley-like structures.

### 2.2. Radon-Like Features

RLF was originated from the idea of Radon transform. The traditional Radon transform was defined as the line integrals of a two-dimensional function f(x,y) along a line l(θ,τ) in the plane, with θ and τ are the slope and intercept of the line. When Radon transform is applied to an image, it will collapse the whole image into a line. Generally, lines with high-intensity values correspond to the bright points, while lines with low-intensity values correspond to the dark points. Therefore, the features can be extracted by using multiple scan lines in different orientations. Since Radon transform performs integral on the whole line, the difference between the regions swept by the line will not be distinguished. In addition, Radon transform is sensitive to scaling, translation and rotation.

Different from traditional Radon transform, RLF will not collapse the image into scalar values via integration of the scan line. Actually, RLF divides the scan line into multiple segments, and then carries out segmental feature extraction along the scan line, so as to better reflect the distribution of features in the image space. In the meantime, when multiple scan lines along various directions are provided, RLF can define a distribution of features. Considering the specific implementation of RLF, two important issues should be resolved: the first issue is related with the segmentation strategy of scan lines. Generally, some edge detection operators (e.g., Canny, Sobel, Kirsch) can be used to provide auxiliary line segments, which means, line segments can be defined by a set of salient points (called ‘knots’) along the scan line. These knots are the intersection points of the scan line and edge map. In this way, the knots provide positive guidance of the constituent structures of the image. The second important issue is related to the extraction function. Supposing the set of knots along a scan line is given as $\left(k_{1},\ldots ,k_{n}\right)$, then, for any one point p that located in the line segment from $k_{i}$ to $k_{i+1}$, the corresponding RLF can be calculated by Equation (Equation 2).
(2)$$\Psi \left(p,l,k_{i},k_{i+1}\right)\left[I\left(x,y\right)\right]=F\left(I,l\left(k\right)\right),\;k\in \left[k_{i},k_{i+1}\right],$$
where I(x,y) denotes the target image, l(k) denotes the *k*-th segment of scan line, and function F(·) is the extraction function. Follow the definition of extraction function, when a series of scan lines with the same slope θ but different intercepts $\left(\tau_{1},\ldots ,\tau_{m}\right)$ were used, the resulting RLF would be a two-dimensional image of the same size as the target image, this is a significant point of departure from the traditional Radon transform where the output in such a case is a one-dimensional vector. Moreover, if the slope θ varied as well, RLF would be presented as a series of feature images.

Here, in order to support the efficiency of the RLF-based feature aggregation scheme, we provided a toy example to illustrate the way of RLF, as shown in Figure 2. First, a bacteria image was shown in Figure 2a, it can be observed that each bacteria body was surrounded by a circle of highlighted areas. Then, the Canny edge detector was performed on the original bacteria image to form an edge map (see Figure 2b), and a series of scan lines with different slopes and intercepts were used to determine knots and line segmentation. For simplicity, we only display three scan lines with 135° slope (red lines) and three scan lines with 45° slope (green lines) in Figure 2b, and the corresponding knots are marked as star-shape. After, a simple form of extraction function was adopted, which calculated the absolute value of the difference between two endpoints (a pair of adjacent knots) of each line segment, and then assigned to all pixels on this line segment. Figure 2c–g illustrated RLF maps obtained by using five groups of scan lines with different slopes, for each group of scan lines, their slopes are equal, while the intercepts are varied and cover the whole image. In addition, Figure 2h illustrated a mean RLF image obtained by averaging the RLF maps of all directions. It can be observed that RLF effectively aggregated image statistics along a line segment, the highlighted areas around each bacteria body were eliminated due to the feature aggregation effect.

## 3. Proposed Method

In this section, we elaborated on our proposed RLF-based FV recognition method. As depicted in [59], RLF has been successfully applied for connectomics image analysis, such as cell boundary enhancement, mitochondria segmentation, and vesicle cluster enhancement, to name a few. However, due to the fact that FV images are generally low-contrast and noisy, it is less effective to perform Radon-like feature extraction directly on the original image. With this in mind, we developed a specific implementation of RLF and performed on the mean curvature images, thus can effectively aggregate dispersed spatially statistics information into compact feature descriptors. After, the extracted Radon-like features would be used for matching and verification. Figure 3 illustrated the block diagram of our proposed FV recognition method. The whole process can be divided into three main steps: first, a robust ROI localization method was performed on the acquired original vein image [62], so as to achieve a more accurately positioned ROI image. Then, the mean curvature of each pixel in the ROI image was calculated, and their corresponding Radon-like features were constructed by selecting eight groups of dense scan lines, which come from eight different directions. After, previously obtained feature images of different directions were accumulated to form a mean RLF image. Finally, after normalization [27,63] and binarization, the resulting binary image would be used for subsequent matching and recognition purposes. It should be noted that, to be fair, we adopted a conventional template-matching algorithm [26] for performance comparison and assessment, the matching ratio between an input pattern and the registered templates was calculated to determine whether to accept or reject. Below, we detail the proposed method step by step.

### 3.1. ROI Localization

The acquired FV images by charge-coupled device (CCD) camera often contain many unexpected background information, which will aggravate the negative impact on the accuracy of vein recognition. Therefore, an effective ROI localization is necessary no matter what feature extraction methods are performed [64]. Here, we adopted a robust ROI localization method that has been proposed in [62]. The main idea of the adopted ROI localization algorithm is to divide the whole FV image into four parts (namely top-left, top-right, bottom-left and bottom-right), and then we carry out a three-level dynamic thresholds strategy on each part of the image, so as to obtain more complete and distinct contour edge information. Finally, the edges from each part of the image are connected to form the finger contour boundaries. In this case, the ROI region is located in the finger contour. Figure 4 illustrated an implementation example of the proposed Radon-like features, among, the ROI localization result corresponding to the Figure 4a was shown in Figure 4b, and more detailed descriptions please refer to [62].

### 3.2. Implementation of Radon-Like Features

As described beforehand, the implementation of RLF contains two important issues, one is related to the segmentation strategy and knots selection, and the other is the form of the extraction function. For the segmentation strategy, we first calculated the mean curvature at each point of the ROI image by using Equation (Equation 1). As observed from Figure 4c, the mean curvature map presented enhanced vein patterns than the ROI image. However, it still contained plenty of break lines, thin lines, as well as pseudo vein patterns. So, we introduced the Canny edge detector to obtain an edge map of the mean curvature image, as shown in Figure 4d. It should be emphasized that the edge detector operation was performed on the mean curvature image rather than the low-contrast original FV image or ROI image. Then, eight groups of scan lines with different slopes and intercepts were intersected with the edge map, so as to obtain the corresponding line segmentations and set of knots. Specifically, the slopes were sampled with 45° intervals from the scope of [0°, 315°]. For each fixed slope value, the intercept values should be guaranteed to cover the whole image. Considering that our purpose was to obtain a vein pattern that is more continuous, genuine, and minimize the influence of pseudo vein information, we specifically designed an implementation form of the extraction function, as defined in Equation (Equation 3).
(3)$$F\left(MC,l\left(k\right)\right)={\frac{\int_{k_{i}}^{k_{i+1}}MC\left(l\left(k\right)\right)\;dk}{∥l\left(k_{i+1}\right)-l\left(k_{i}\right)∥}}_{2},\;k\in \left[k_{i},k_{i+1}\right],$$
where MC(x,y) was the mean curvature image, which indicated the processing target of RLF. l was a scan line along which the RLF was calculated. The numerator of the extraction function (Equation 3) indicated the piecewise integral along a scan line in the mean curvature image, and the denominator of the extraction function (Equation 3) was the distance of two knots in each line segment. In this manner, the extraction function (Equation 3) can capture the most dominant response at each pixel by assigning an equal value to all pixels between the knots $k_{i}$ and $k_{i+1}$ along scan line l. The corresponding result RLF image was shown in Figure 4e,f, among, Figure 4f was the pixel-wise mean of RLF accumulated from eight different directions. As compared with the mean curvature image, the mean RLF image further enhanced the vein patterns, and the vein network becomes more continuous, the related line width information were also restored, thus leading to a more smoothing vein structure image. It is due to the fact that the RLF implementation can effectively aggregate the dispersed spatially statistics information into compact feature descriptors, thus further highlight the vein patterns and suppress spurious non-boundary responses and noises.

### 3.3. Template Matching

After finishing the aggregation of RLF, we can obtain a smooth vein structure image, which would be used for subsequent matching and recognition. Here, we adopted a conventional template-matching method for fair assessment [26], which has shown robustness to the shifting of matching images. The matching process was carried out by searching for an optimal overlapped region of the registered template image and the input image. As shown in Figure 5, supposing R(x,y) and I(x,y) are the registered and input matching images, respectively, and *w* and *h* are the width and height of both images. Considering the displacement, two margins from the registered image, that denoted as $c_{w}$ and $c_{h}$, were cut to obtain the registered template sub-image. (In the following experiments, we discussed the parameter setting of $c_{w}$ and $c_{h}$). As a result, the template data was determined by the red rectangular region within R(x,y) (as shown in Figure 5a). Then, the template window slid from the top-left corner of the input image (green window in Figure 5b), so as to find the optimal matching position, which means that the template data and the input data has the maximum overlapped region in this position. At this point, we can give the formula of match rate, as shown in Equation (Equation 4):(4)$$R_{m}=\frac{N_{common}}{\left(N_{template}+N_{input}\right)}$$
where the numerator $N_{common}$ represents the number of matched pixel pairs when the registered template sub-image and the input image region have reached the optimal match. $N_{template}$ and $N_{input}$ are the number of pixels in the maximum overlapped region of the template image and the input image, respectively. $R_{m}$ is the match ratio. Obviously, when the template region is exactly matched with the input region, $R_{m}=0.5$, while when the number of pixels of the overlapped region is zero, $R_{m}=0$, that means the registered finger is completely different from the input finger. Thereby, the ratio value of $R_{m}$ is in a range of 0 to 0.5, a larger value means a better match, while a smaller value means a poorer match. If the value of $R_{m}$ is larger than a preset threshold value, it will be accepted, otherwise, it will be rejected.

For clarity, we provided a comparison of the match ratio by using the mean curvature and the proposed RLF-based method from the perspectives of intra-class and inter-class, as shown in Figure 6 and Figure 7, respectively. In Figure 6, the first row shows the extracted vein patterns by using the mean curvature method, and the second row shows the extracted vein patterns by using the proposed RLF-based method. The first column is the registered finger, the second and the third columns are two different input images from the same finger class. Below the image, the corresponding match ratios with the registered template image are also presented. If we use the registered image as the input for matching, the match ratio is 0.5, since they are the same image. As observed from Figure 6, the proposed RLF-based method achieved higher match ratios than the mean curvature. This is due to the fact that the aggregation of RLF is able to retrieve more ignored structure characteristics, e.g., the growth direction and varied width, which may be helpful in vein pattern representation and matching.

For assessing the results of inter-class, we randomly choose two input images from different finger classes, as shown in the second and third columns of Figure 7. Although the setting is almost the same as in Figure 6, the calculated match ratio of both methods are low, and the match ratios of mean curvature are even lower than the proposed RLF-based method, it is because the proposed RLF-based method enhanced the vein patterns, so we can get more overlapped points even though the two matching images are derived from different finger classes.

## 4. Experimental Analysis

To ascertain the effectiveness of our proposed RLF aggregation-based FV recognition method, we carried out comprehensive experiments on four different FV databases which were constructed by using different sensors. First, a brief description of the adopted FV databases was provided in Section 4.1. Then, the experimental setting and assessment criteria were reported in Section 4.2. Next, in Section 4.3, in order to objectively evaluate the matching performance of the proposed RLF-based method, we conducted experimental analysis on two margin parameters of $c_{w}$ and $c_{h}$, which are used in the template matching algorithm to determine the registered template sub-image. After, in Section 4.4 and Section 4.5, the recognition performance of the proposed method was analyzed from the perspectives of quantitative and visual observation, respectively. Lastly, the computational time of main steps in our proposed RLF-based method was measured and compared in Section 4.6. In addition, it should be noted that we carried out all of the experiments under a computing environment with 3.6 GHz Intel Core i7 CPU and 32 GB RAM.

### 4.1. Finger Vein Databases

Table 1 shows the relevant properties of the four FV databases used in our experiments. Among these, the first three databases are publicly available, hereinafter named as ‘HKPU’ [65], ‘MMCBNU’ [66], and ‘FV-USM’ [67], respectively. Moreover, in order to verify the effectiveness of a large FV image database, a new database (namely ‘ZSC-FV’) was collected at the University of Electronic Science and Technology of China, ZhongShan Institue. The ‘ZSC-FV’ database contains 1030 volunteers, all are college students with ages ranging from 18 to 22 years old. Each individual provided 6 image samples from the index, middle and ring fingers of both hands, thus a total of 36 finger samples for each individual, and a total of 37,080 FV image samples. The whole collection process was carried out under varying indoor lighting conditions, some indoor positions were illuminated by strong spotlight sources, and some indoor positions were mainly illuminated by ambient lights. All acquired original FV images are in 8-bit bitmap format with 256 grayscale levels, and have the same size of 384×512. The acquisition equipment is EA, manufactured by Beijing Yannan Tech Co., Ltd., Beijing, China. The fingertip is oriented to the right and outside the image region.

### 4.2. Experimental Settings and Assessment Criteria

#### 4.2.1. Experimental Settings

As observed from Table 1, Different FV databases own different size and quality of image samples, and the preserved background scopes are also diverse. In this regard, we should do some cropping and resizing so as for further use.

In the ‘HKPU’ database [65], most image samples contain a rectangle frame, as well as serious shadow interfere, we cut 30 pixels at the top boundary, 10 pixels at the bottom boundary, 30 pixels at the left boundary, and 50 pixels at the right boundary. Then, we resized the cropped images to half-size by bicubic interpolation, thus obtaining the final image samples with a size of 109×217 (as shown in the last row of Table 1).

The ‘MMCBNU’ database [66] has a relatively clean and pure black background, so we only cut out an area of five pixels at four boundary positions. Then, we resized the cropped images to one-quarter size, thus obtaining the final image samples with a size of 118×158.

Unlike other FV databases, the image samples in the ‘FV-USM’ database [67] are fingertip downward and contain plenty of useless information. Therefore, we first rotated the images by $90^{{}^{\circ}}$, then, we cut out 150 pixels at the top and bottom boundaries, respectively, 5 pixels at the left boundary, and 70 pixels at the right boundary. After, we resized the cropped images to half size, thus obtaining the final image samples with a size of 171×203.

‘ZSC-FV’ database [62] also has a very complicated background and high edge density in the noisy regions. Therefore, we cut out an area of 20 pixels at four boundary positions. Then, we resized the cropped images to half size by bicubic interpolation, thus obtaining the final image samples with a size of 173×237.

#### 4.2.2. Assessment Criteria

In order to quantitatively assess the matching performance of our proposed method, we adopted some typical measurement criteria in the experiments, as detailed below:False Acceptance Rate (FAR), it is the error rate where the un-enrolled FV images are accepted as enrolled images. The related formula is shown in Equation (Equation 5).
(5)$$FAR=\frac{NFA}{NIRA}\times 100\%,$$
where NFA is the number of false accept, and NIRA is the number of impostor recognition attempts.False Rejection Rate (FRR), it is the error rate where the enrolled FV images are rejected as un-enrolled images. The related formula is shown in Equation (Equation 6).
(6)$$FRR=\frac{NFR}{NGRA}\times 100\%,$$
where, NFR is the number of false reject, and NGRA is the number of genuine recognition attempts. Taking each finger as one class, if there are *n* number of finger classes, and each finger class has *m* number of images. NGRA will be n×m, and NIRA will be (n−1)×m.Equal Error Rate (EER), it is defined as the ratio of trials in which the FAR is equal to the FRR. However, there may not exist a threshold such that FAR is exactly equal to FRR in practice, because FAR and FRR are both discrete values. In this case, we adopted an approximate calculation method for EER. Concretely, the EER is calculated as follows: First, let *T* be a set of threshold values, which are sampled from 0 to 0.5 (since the match ratio is in the range of [0,0.5]) with a sampling interval of 0.0001, namely T={0,0.0001,0.0002,…,0.5}. In this case, there are 5001 elements in set *T*. Supposing $T_{i}$ is the *i*-th threshold of *T*, with i={1,2,…,5001}. If the match ratio is lower than the predefined threshold $T_{i}$, the claimant will be accepted, otherwise, the claimant is rejected. Therefore, we can obtain a couple of $FAR_{i}$ and $FRR_{i}$ for each threshold $T_{i}$. When the threshold $T_{i}$ is varied from 0 to 0.5, the corresponding $FAR_{i}$ will be reduced and $FRR_{i}$ will be increased. Finally, the EER can be obtained by calculating $(FAR_{i}+FRR_{i})/2$ when $∥FAR_{i}+FRR_{i}∥$ is minimized.

### 4.3. Analysis on the Margin Parameters

As depicted in Section 3.3, in order to eliminate the effect of image shifting, part of the horizontal and vertical margin areas in the registered images need to be cut out, so as to facilitate the search of the optimal matching region in the input image. In this experiment, we analyzed the matching performance of the proposed RLF-based method under different margin parameter values of $c_{w}$ and $c_{h}$, which are used to crop the template sub-image from the registered image. It should be noted that registered images derived from different FV database’s own diverse image sizes, as well as with different background areas retained, thus the values of $c_{w}$ and $c_{h}$ will be affected by these factors. With this consideration, we tested six groups of different margin parameter values on four FV databases, covering the range from $c_{w}$ = 5, $c_{h}$ = 5 to $c_{w}$ = 50, $c_{h}$ = 50.

In addition, considering that some FV databases have provided a built-in ROI result image set, we also choose two sources of ROI images for parameter analysis. One is derived from our adopted ROI localization strategy [62] (see Section 3.1), and the other is the built-in ROI images provided by some public FV databases. It should be clarified, compared with the ROI images obtained by our method, the ROI images provided by those publicly available FV databases only contain a small part of the whole finger region (mainly concentrated in the middle of the finger region), which means that the contour of the finger is lost and the correction of finger placement becomes impossible. To illustrate this point, we presented two sample diagrams from MMCBNU and FV-USM databases respectively, as shown in Figure 8 and Figure 9. As can be observed from these samples, the built-in ROI images generally have a relatively smaller size than our extracted ROI images, in MMCBNU database, the size of built-in ROI image is 60×128, while our extracted ROI images have a size of 118×158. Likewise, in FV-USM database, the size of the built-in ROI image is 100×300, while our extracted ROI images have a size of 171×203. Furthermore, the ROI images obtained by our method still retain a small part of background information, while the built-in ROI images only contain a part of finger vein regions. In this case, we also hope to explore whether the retained background information has a positive or negative influence on the matching results.

Table 2 illustrated the EER results of the RLF-based matching with different margin parameters on four FV databases. As we can observe, some EER results are missing because of the size of the image, for example, in case of $c_{w}$ = 50, $c_{h}$ = 50, most of the EER results of the built-in ROI are missing due to their small image size. At the same time, we can draw some conclusions: firstly, there are no unique and fixed-parameter values of $c_{w}$ and $c_{h}$ that can satisfy all the situations, aims to different size of ROI images, the optimal margin parameters are different. Secondly, compared with the built-in ROI, our extracted ROI preserved a complete finger contour and part of the background, which can provide better auxiliary information about finger placement, and help to find more accurate template matching region.

To sum up, in the subsequent experiments, we will select two groups of margin parameters ([$c_{w}$ = 30, $c_{h}$ = 30] and [$c_{w}$ = 40, $c_{h}$ = 40]) to compare the effectiveness of different feature extraction algorithms.

### 4.4. Quantitative Comparison of Matching Performance

In this experiment, the matching recognition performance of our proposed method on four databases was quantitatively analyzed. Each finger is taken as one class, and all the captured image samples from the same finger belong to the same finger class.

For comparison, we also provided the EER results of five unsupervised handcrafted feature extraction methods (including LLBP [12], Gabor [60], WLD [27], maximum curvature [28], mean curvature [30]), and one newly developed CNN-based method, hereinafter we called it the ‘GaborPCA’ network [48]. Similar to the aforementioned handcrafted counterpart methods, the GaborPCA network also uses an unsupervised fashion and no class label information is needed in the training procedure. The GaborPCA network has a 3-layer CNN architecture with two convolutional layers and one binarization layer, in which, the first convolutional layer is tuned by using PCA filters, and the second convolutional layer is tuned by using adaptive Gabor filters.

For those compared handcrafted-based methods, since different threshold values may lead to quite different results, we uniformly adopted the Otsu threshold strategy [68] to binarize the extracted vein pattern images. In addition, for the template matching issue, we presented the EER results under two different settings of margin parameters of $c_{w}$ and $c_{h}$. While for the GaborPCA network, the outputs were all 1D feature vectors, therefore, we adopted the Euclidean distance to calculate the match results, and then, the corresponding match results were used to calculate the FAR, FRR and EER in the same way as shown in Section 4.2.2.

Moreover, in this experiment, the usage of databases is also different between the handcrafted-based methods and GaborPCA network. For our proposed RLF-based method and five handcrafted-based methods, they do not need a training procedure, so only the testing settings are required. Specifically, for the HKPU database [65], all of 312 finger classes that derived from a total of 156 individuals in Session 1 were used for testing, and each finger contributed 6 images, bringing the total number of images to 1872. For the MMCBNU database [66], considering that the number of finger classes will affect the EER results, we randomly choose 312 finger classes for testing, and each finger randomly contributed 6 number of image samples, bringing the total number of images to 1872. The experiments are repeated five times. The same experimental settings are also used in FV-USM [67] and ZSC-FV [62] databases, and the experiments are repeated five times in FV-USM and ten times in ZSC-FV, since ZSC-FV is a bigger one. Finally, the average results of several experiments are reported.

While for the GaborPCA network, the detailed settings of training and testing procedures are shown below: For the HKPU database [65], 210 number of finger classes with a total number of 1260 images (six for each class) were used for training, these image samples are acquired from Session 2. Then, the same settings as before were used for testing, which means that there is a total number of 1872 image samples with 312 finger classes. For the MMCBNU database [66], 288 finger classes with a total number of 2880 images (10 for each class) were randomly chosen for training, then, the remaining 312 finger classes with a total number of 1872 images (randomly choose 6 for each class) were used for testing. For the FV-USM database [67], 180 finger classes with a total number of 2160 images (12 for each class) were randomly chosen for training, then, the remaining 312 finger classes with a total number of 1872 images (randomly choose six for each class) were used for testing. For the ZSC-FV database [62], 5868 finger classes with a total number of 35,208 images (6 for each class) were randomly chosen for training, then, the remaining 312 finger classes with a total number of 1872 images (randomly choose six for each class) were used for testing.

The FAR-FRR curves are shown in Figure 10 and Figure 11, and their corresponding EER results are shown in Table 3 and Table 4. Among these, a smaller EER value denotes the better of method. Moreover, the EER values are also affected by the number of finger classes. The smaller the number of finger classes, the smaller the EER value. The experimental results show that, on the HKPU database, the GaborPCA network obtained the worst result. While for the MMCBNU, FV-USM and ZSC-FV databases, Gabor filter produced a worse result. Finally, for the GaborPCA network and our proposed RLF-based method, very close and promising results were obtained on all the databases except for the HKPU database, this is due to the fact that the HKPU database has fewer training samples than the other three databases. All in all, the experimental results further confirmed that our proposed method has robustness to the threshold selection.

### 4.5. Visual Assessment of Matching Performance

In this experiment, we visually assessed the extracted FV features of various methods, so that we can get more insights into the proposed RLF-based method. Figure 12 shows the extracted FV features that were originated from five commonly used methods (including LLBP [12], Gabor filter [60], WLD [27], maximum curvature [28], and mean curvature [30]), as well as our proposed RLF-based method. Since the outputs of the GaborPCA network are 1D feature vectors, we do not show this here. It should be emphasized that, in order to obtain optimal results, different methods may use different threshold values. However, for the sake of fair comparison, we still adopted the same Otsu threshold strategy [68] in the following experiments. Some specific issues can be observed in Figure 12, as detailed below:In the third row of Figure 12, though the LLBP [12] method extracted more smooth and continuous vein patterns, it also introduced plenty of pseudo vein points.In the fourth row of Figure 12, the adopted Gabor filters [60] contained three scales (wavelength is set to 16, 17, 18) and eight orientations (from 22.5° to 180° with equal intervals), thus a total of 24 filters. The final result was obtained by taking the minimum value of all filters. However, the results seem poor, which is due to the fact that the method of the Gabor filter is sensitive to the threshold values, maybe a different threshold value would bring a better result.In the fifth row of Figure 12, there is a lot of noise in the result of the WLD method [27], and the extracted vein patterns are very discontinuous.In the sixth row of Figure 12, similar to the WLD method, the maximum curvature-based method [28] still missed a lot of vein information under the Otsu threshold strategy.In the seventh row of Figure 12, although the mean curvature method [30] extracted more complete vein patterns, it still has the problem of discontinuity of vein lines.Finally, as shown in the last row of Figure 12, our proposed RLF-based method obtained more continuous vein lines, that is, some breaking points, which existed in the result of the mean curvature method, have been connected, thus obtain more complete and enhanced vein patterns.

To sum up, compared with some other FV feature extraction methods, our RLF-based method can obtain more complete and continuous vein patterns as well as better noise resistance.

### 4.6. Time Analysis

In the last experiment, we provided a measurement of the computational time of the main steps in our proposed RLF-based method. For comparison, we also provided the time costs of the other methods. It should be noted that, for our proposed RLF-based method, the recorded time cost mainly covered the procedure of feature extraction on the ROI image, which means, the calculation of mean curvature image, the Canny edge map, and the Radon-like feature images, are all covered, while some preprocessing steps of cropping and ROI localization, and postprocessing steps of normalization and binarization, are not mentioned in the recorded times. For the other compared methods, we also only recorded the time cost of the corresponding feature extraction step. Table 5 shows the computational times (in milliseconds) of various methods on four FV databases. Regardless, the mean curvature method achieved the shortest time cost, this is because the mean curvature is a kind of two-dimensional curvature operator in nature, thus can directly perform the calculation on the image. While for the maximum curvature, it is a one-dimensional curvature operator, which has to be performed on each cross-sectional profile. Likewise, the LLBP is a global (image-level) feature extraction method, it needs to be calculated in the neighborhood space of each pixel, thus leading to a huge amount of computation burden. We have to admit that our proposed RLF-based method requires more time than the mean curvature method, as a result of some additional steps are introduced, especially for the decision of the knots and the execution of extraction function in the Radon-like feature extraction step. In spite of these reasons, our proposed method shows better than the LLBP and maximum curvature methods. On the whole, the time cost of our proposed method is acceptable.

## 5. Conclusions

In this paper, we carried out an exploration on the aggregation ability of RLF in the field of FV recognition, and proposed a novel FV feature extraction method. The proposed method combined the mean curvature and RLF, which can effectively aggregate the dispersed spatial information around the vein structures. As a result, the vein patterns can be highlighted, and spurious non-boundary responses and noises can be suppressed. Finally, a more smoothing vein structure image can be obtained. The experimental results on four FV databases confirmed the superiority of our proposed method, and compared with some state-of-the-art FV recognition methods, our proposed method can not only preserve the intrinsic vein patterns, but eliminate most of the pseudo vein information, leading to more smoothing and a genuine vein structure image. As with any new method, there still have some unresolved issues that deserve further consideration. First, for the specific implementation of RLF, we presented a relatively simple form of extraction function, and achieved good performance. However, whether there exist some more efficient forms of extraction function, deserves further investigation. Second, we must point out that, even though a series of RLF images were obtained by using our method, only a mean RLF image was used in the experiments. Further studies are needed to clarify whether there have other forms of feature fusion strategies. Third, for the calculation of line segmentation and knots, we adopted a serial form to intersect the edge map with each scan line in turn. In this case, the computational speed can still be improved. In the future, we will try to convert the serial implementation into parallel implementation, which will use parallel programming techniques to synchronously calculate the intersection of all scan lines with the edge map.

## Figures and Tables

**Figure 1 sensors-21-01885-f001:**
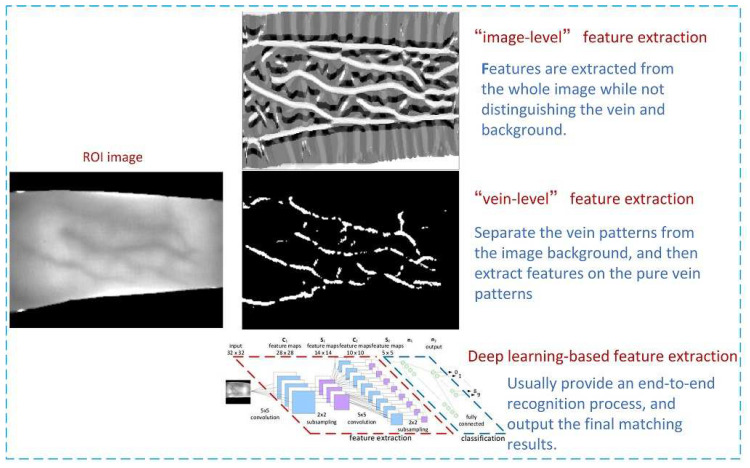
Illustration of different ways of feature extraction for finger vein images.

**Figure 2 sensors-21-01885-f002:**
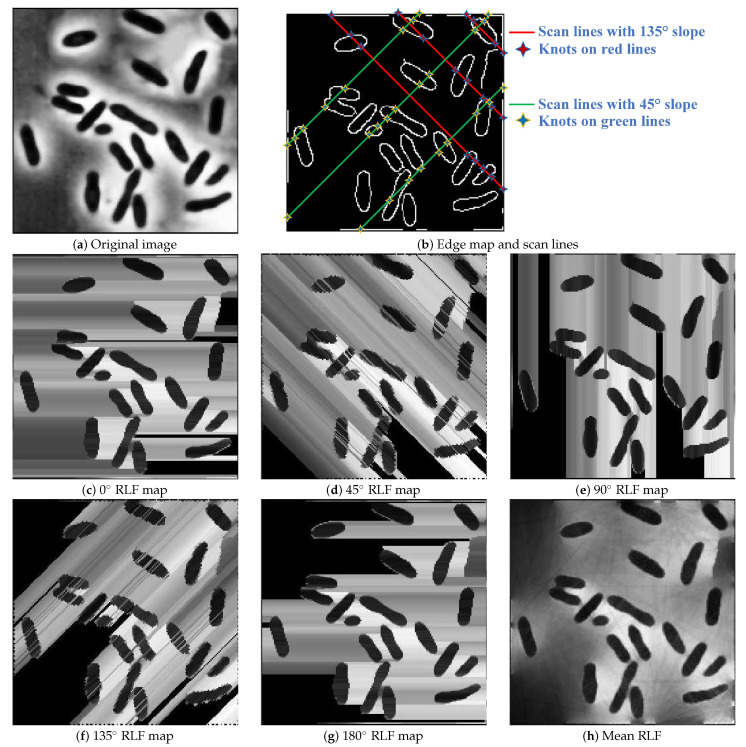
Toy example: illustration of the basic principle of radon-like features (RLF). (**a**) Original bacteria image; (**b**) edge map obtained by using Canny edge detector, this figure also illustrated the line segments and knots; (**c**) 0° RLF map; (**d**) 45° RLF map; (**e**) 90° RLF map; (**f**) 135° RLF map; (**g**) 180° RLF map; (**h**) mean RLF map obtained by averaging RLF maps of all directions.

**Figure 3 sensors-21-01885-f003:**
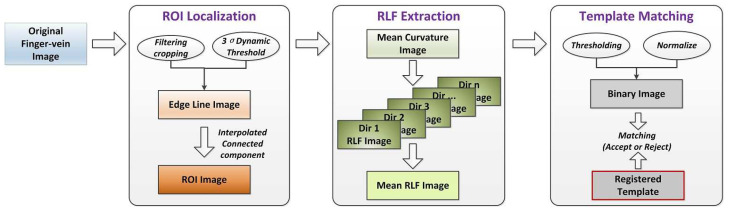
Block diagram of the proposed RLF-based feature extraction method for finger vein image recognition.

**Figure 4 sensors-21-01885-f004:**
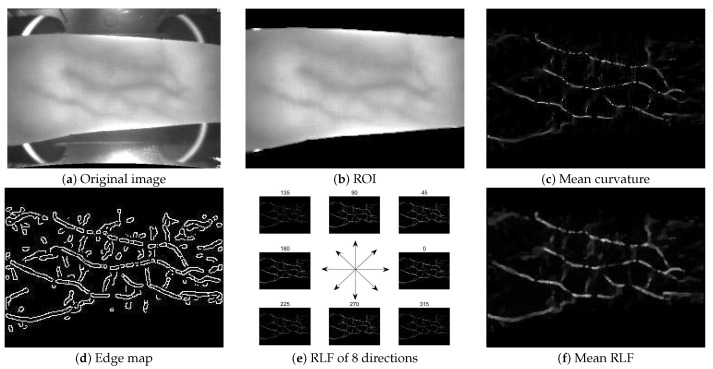
An implementation example of the proposed Radon-like features. (**a**) Original finger vein (FV) image; (**b**) region of interest (ROI) result; (**c**) mean curvature result; (**d**) edge map of mean curvature image; (**e**) scan lines of eight directions; (**f**) mean RLF image.

**Figure 5 sensors-21-01885-f005:**
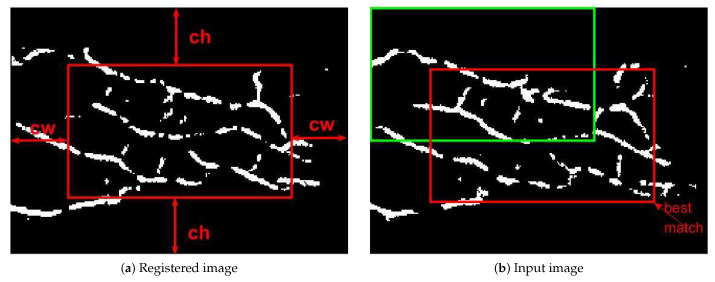
Template matching between the registered and input images. (**a**) Cut $c_{w}$ and $c_{h}$ from the registered image margin; (**b**) the best match between the registered and input images, the match ratio is 0.259.

**Figure 6 sensors-21-01885-f006:**
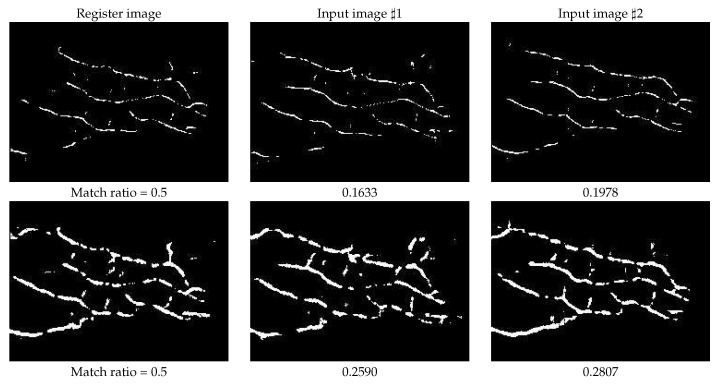
The match ratio of intra-class, that means the registered template and input image are both from the same finger class. The first row shows the results of mean curvature method, and the second row shows the results of our proposed RLF-based method. Their corresponding match ratios are listed below the images.

**Figure 7 sensors-21-01885-f007:**
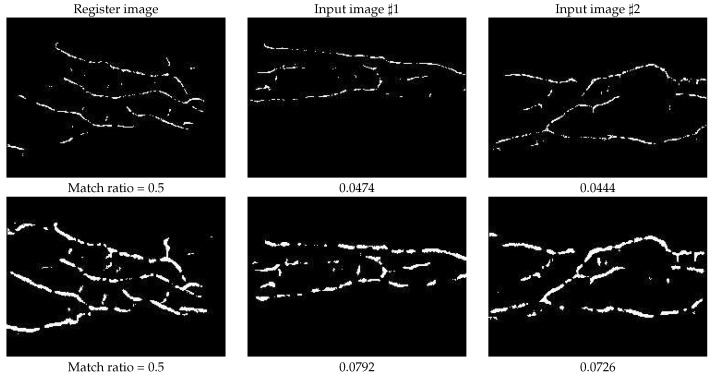
The match ratio of inter-class, that means the registered template and the input image are from different finger classes. The first row shows the results of mean curvature method, and the second row shows the results of our proposed RLF-based method. Their corresponding match ratios are listed below the images.

**Figure 8 sensors-21-01885-f008:**
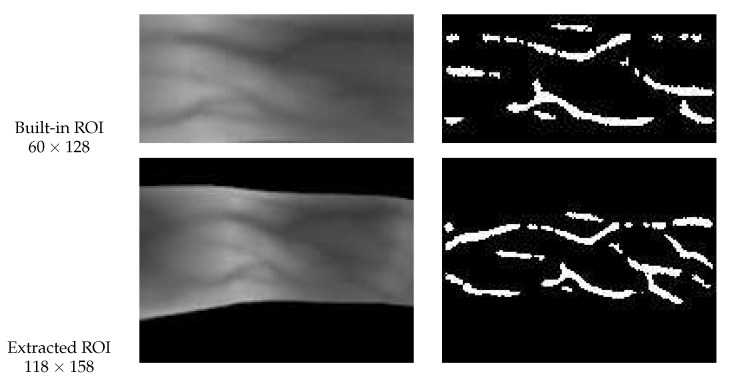
Sample diagram from MMCBNU database, the first row is the built-in ROI image and its corresponding RLF-based vein feature, while the second row is derived from our adopted ROI localization strategy [62] and its corresponding RLF-based vein feature.

**Figure 9 sensors-21-01885-f009:**
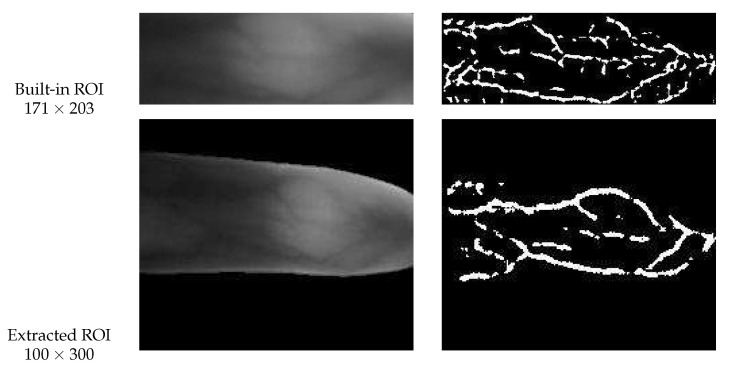
Sample diagram from FV-USM database, the first row is the built-in ROI image and its corresponding RLF-based vein feature, while the second row is derived from our adopted ROI localization strategy [62] and its corresponding RLF-based vein feature.

**Figure 10 sensors-21-01885-f010:**
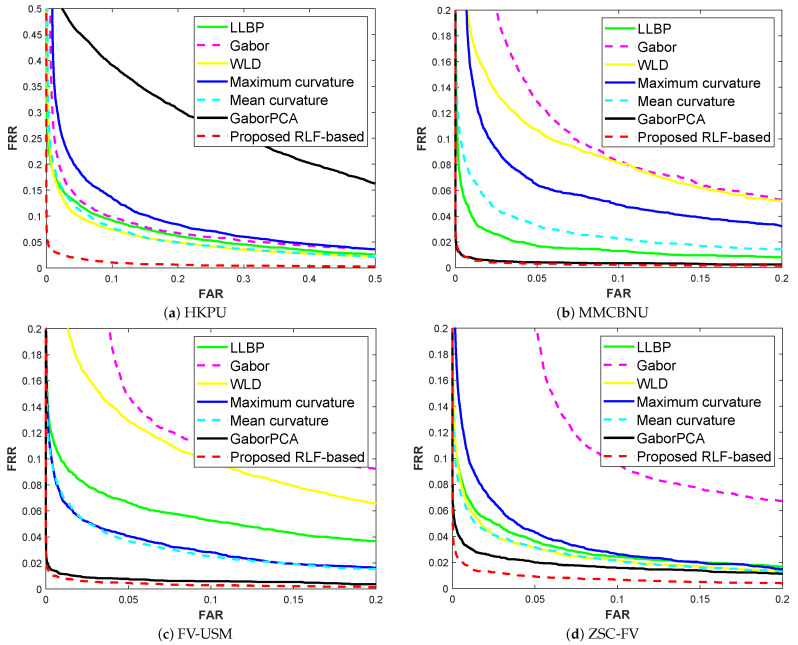
False acceptance rate (FAR)-false rejection rate (FRR) curves of compared methods on four finger vein databases, the margin parameters are $c_{w}$ = 30, $c_{h}$ = 30.

**Figure 11 sensors-21-01885-f011:**
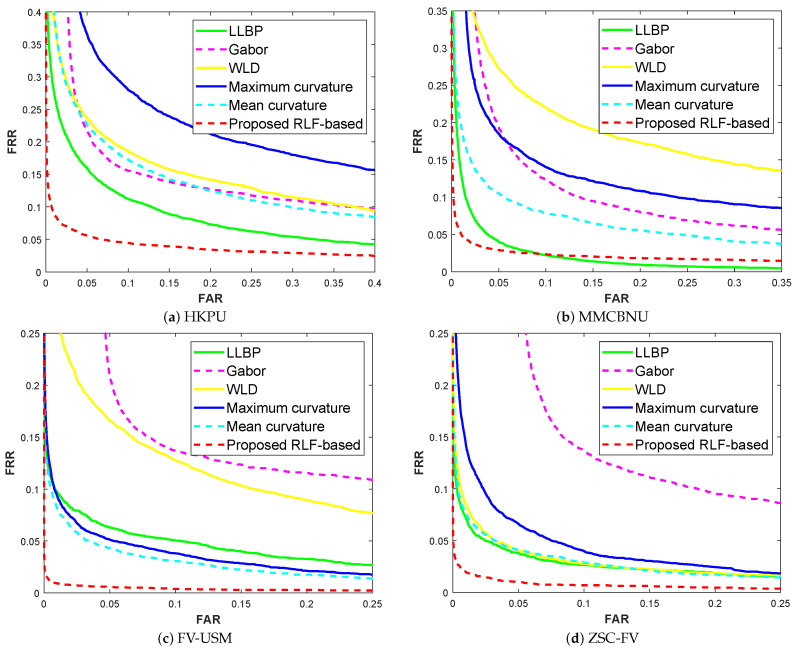
FAR-FRR curves of compared methods on four finger vein databases, the margin parameters are $c_{w}$ = 40, $c_{h}$ = 40.

**Figure 12 sensors-21-01885-f012:**
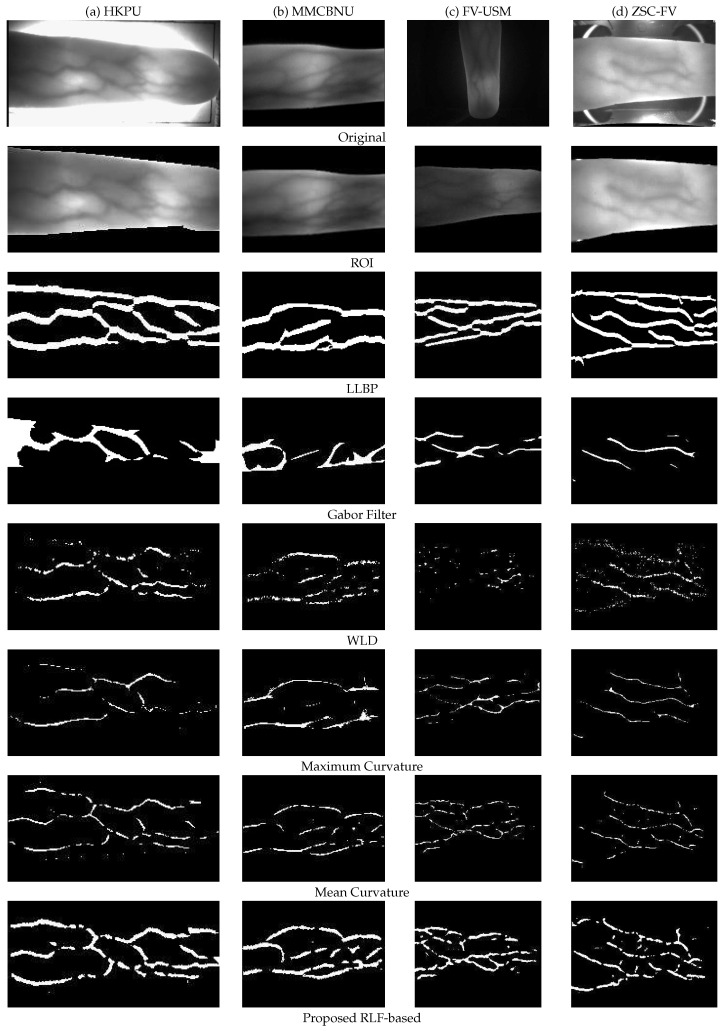
Different finger vein feature extraction methods are carried out on four image samples from different databases. The first and second rows show the originally acquired images and their corresponding ROI images. Here, we uniformly adopted the strategy in [62] to obtain ROI. The third to eighth rows show the extracted feature images by using LLBP, Gabor Filter, WLD, Maximum Curvature, Mean Curvature, and our proposed RLF-based method, respectively.

**Table 1 sensors-21-01885-t001:** Details of four finger vein databases.

Databases	HKPU	MMCBNU	FV-USM	ZSC-FV
Num of individuals	156	100	123	1030
Fingers	index, middle	index, middle, ring	index, middle	index, middle, ring
Hands	left	left, right	left, right	left, right
Num of images per finger	6/12	10	12	6
Sessions	2	1	2	1
Num of finger classes	312	600	492	6180
Total num of images	3132	6000	5904	37,080
Image size	513×256	480×640	640×480	384×512
Scaled image size	109×217	118×158	171×203	173×237

**Table 2 sensors-21-01885-t002:** Equal error rates (EER) obtained by using different margin parameters on four finger vein databases.

	HKPU		MMCBNU		FV-USM		ZSC-FV
	Built-In	Extracted	Built-In	Extracted	Built-In	Extracted	Extracted
	ROI	ROI	ROI	ROI	ROI	ROI	ROI
$c_{w}$ = 5, $c_{h}$ = 5	–	–	**2.12%**	–	4.28%	–	–
$c_{w}$ = 10, $c_{h}$ = 10	21.12%	2.55%	2.36%	1.60%	1.93%	**0.74%**	2.02%
$c_{w}$ = 20, $c_{h}$ = 20	10.60%	**2.28%**	18.82%	**0.77%**	**1.68%**	0.76%	1.43%
$c_{w}$ = 30, $c_{h}$ = 30	5.90%	2.49%	–	0.78%	5.56%	0.87%	**1.39%**
$c_{w}$ = 40, $c_{h}$ = 40	4.72%	5.47%	–	0.93%	26.87%	0.93%	1.69%
$c_{w}$ = 50, $c_{h}$ = 50	**4.23%**	–	–	–	–	–	2.32%

**Table 3 sensors-21-01885-t003:** EERs obtained by using different methods on four finger vein databases, the margin parameters are $c_{w}$ = 30, $c_{h}$ = 30.

Databases	LLBP	Gabor Filter	WLD	Maximum Curvature	Mean Curvature	GaborPCA	Proposed RLF-Based
HKPU	9.39%	9.82%	8.04%	12.02%	8.56%	26.7%	**2.49%**
MMCBNU	2.59%	9.01%	8.69%	5.99%	3.79%	0.84%	**0.78%**
FV-USM	6.16%	10.76%	9.89%	4.32%	4.08%	1.14%	**0.87%**
ZSC-FV	4.06%	9.76%	3.62%	4.55%	3.63%	2.47%	**1.39%**

**Table 4 sensors-21-01885-t004:** EERs obtained by using different methods on four finger vein databases, the margin parameters are $c_{w}$ = 40, $c_{h}$ = 40.

Databases	LLBP	Gabor Filter	WLD	Maximum Curvature	Mean Curvature	Proposed RLF-Based
HKPU	10.86%	14.14%	15.6%	20.87%	14.64%	**5.47%**
MMCBNU	4.46%	11.29%	17.85%	12.81%	8.39%	**3.3%**
FV-USM	5.99%	12.87%	11.79%	5.10%	4.51%	**0.93%**
ZSC-FV	4.11%	12.28%	4.37%	5.93%	4.37%	**1.69%**

**Table 5 sensors-21-01885-t005:** Computational times (ms) of various methods on four finger vein databases.

Databases	Image Size	LLBP	Gabor Filter	WLD	Maximum Curvature	Mean Curvature	Proposed RLF-Based
HKPU	109×217	254.6	72.3	39.7	231.2	**4.4**	104
MMCBNU	118×158	141.8	40	30.1	182.3	**3.4**	66.7
FV-USM	171×203	232.3	105.2	58	343.9	**5.0**	102.1
ZSC-FV	173×237	377.4	96	70	400.6	**6.5**	143

## Data Availability

Not applicable.

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
