# Peer review of "A Novel Finger Vein Recognition Method Based on Aggregation of Radon-Like Features"

_sensors, 2021, doi:10.3390/s21051885_

Round 1
Reviewer 1 Report
This manuscript proposed a novel finger vein feature extraction method based on the combination of curvature and radon-like features. The idea of applying the RLF for FV feature representation is novel, and the experimental results on four FV databases demonstrated the effectiveness of the proposed method. However, the authors need to clarify the followings in order to improve the manuscript:
- In section 1. Introduction, the authors reviewed that PCANet [39] and Convolutional Autoencoder [40] both belong to lightweight network architectures. I think it's inappropriate since they both have complex architectures, so I suggest to classify them as more powerful but complex network models.
- In section 3.2. Implementation of Radon-like Features, the authors claimed that “eight groups of scan lines with different slopes and intercepts were intersected with the edge map, so as to obtain the corresponding line segmentations and set of knots.”. I suggest to provide more detailed descriptions about how to set the slopes and intercepts of these scan lines, since they determine the number of scan lines actually used for calculating RLF.
- In Table 2,3,4, some EER results are presented, but the authors should clarify whether the unit of measurement of EER is %?
Reviewer 2 Report
The objective of this work is clear, and the presentation is easier to follow.
However, the quality of the paper can be further enhanced, and the comments are listed for authors:
- There are so many acronyms in the manuscript, and the full names of several acronyms are missed, such as ROI, CCD, etc. Please improve.
- As for the experimental settings, since different databases have diverse sizes and qualities, cropping is required and performed. It is a reasonable operation, while, how to define the numbers of pixels cut on each boundary of the images? For example, in the case of the ZSC-FV database, an area of 20 pixels at four boundaries has been cut off. So, why consider 20 pixels, and what happened if consider 15 or 25? Does it affect the result of matching in the recognition?
- Regarding the template matching, how to consider the registered image or the template? For example, if the number of images per finger is 6, does it utilize one of them to be the template, and then the rest five are used as the testing data? If so, how to settle the template among the six images? Please clarify.
- Several threshold values have been employed in this work. For example, in the quantitative comparison of matching performance, the threshold values applied in different methods are derived from the Otsu threshold strategy. Nonetheless, the corresponding values have not been presented. Can the authors provide the details about the threshold values used? Those values could be the references that are helpful for relevant studies.
- As seen in Figure 4, for acquiring the RLF features, the authors considered 8 directions with each of 45°. Besides, the time cost of the proposed method is not optimal among the comparisons. Thus, can the computation cost save when the number of directions decreases? If so, what is the trade-off between the number of the directions and performance? Can the time issue be improved by considering the process of the feature extraction?
- About the contributions, the authors declared that the implemented RLF-based feature extraction method demonstrated a fast running speed and a relatively low complexity of the algorithm. However, as concluded, the computational speed is needed to be improved. Also, in the comparisons, the proposed method shows limitations in the time analysis. Please elaborate them more.
Reviewer 3 Report
In this manuscript, authors proposed a novel finger vein recognition method. It combines curvature and radon-like features. In addition, the experimental results showed that the proposed method can acquire complete and continuous vein patterns with high accuracy. This paper is well organized and the description of the proposed method is straightforward. However, to improve the quality of this paper, there are some issues that should be addressed.
- There is doubt as to whether the performance comparison targets are correct. The conventional methods compared in this paper are relatively old. I think it is necessary to compare it with the latest methods.
- I would like to observe the effect of the proposed method with more image data. There is only one image data in the manuscript. Thus, it is recommended to supplement more comparison images.
Reviewer 4 Report
The Authors present a novel feature extraction method of finger vein (FV) images. The approach combines curvature and radon-like features (RLF) which highlights vein patterns, suppresses spurious non-boundary response sand noises, and results in more smoothed vein structure images. Exhaustive experiments with four FV databases (three publicly available) confirm the effectiveness of the method in comparison with known approaches.
The contributions are clearly described, the method, experimental results, and future research lines are convincing. The paper is well written and it is worth publishing.
Remarks:
- I would use present tense, at least in sentences describing the contributions and the organization of the paper.
- Reference [60]: please correct the bibliographic data.
Round 2
Reviewer 2 Report
The authors corrected the paper to some extents, but there are still grammatical issues and typos in the context. Please read the text carefully and make the corrections. For example, please avoid long sentence, such as in the last paragraph, “such as kernel function-based fusion strategy, can bring good results deserves further study.”
Besides, regarding the future work, could you please briefly clarify how the parallel programming improves the efficiency of the proposed RLF extraction algorithm
Reviewer 3 Report
The issues I mentioned have been well supplemented and revised. I recommend the publication of the manuscript as it is.
Author Response
We gratefully thank the reviewer again for his/her careful reading.